# Self-Penetrating Oligonucleotide Derivatives: Features of Self-Assembly and Interactions with Serum and Intracellular Proteins

**DOI:** 10.3390/pharmaceutics15122779

**Published:** 2023-12-14

**Authors:** Irina Bauer, Ekaterina Ilina, Timofey Zharkov, Evgeniya Grigorieva, Olga Chinak, Maxim Kupryushkin, Victor Golyshev, Dmitry Mitin, Alexey Chubarov, Svetlana Khodyreva, Elena Dmitrienko

**Affiliations:** 1Institute of Chemical Biology and Fundamental Medicine SB RAS, 630090 Novosibirsk, Russia; i.bauer@g.nsu.ru (I.B.); timazharkov74@gmail.com (T.Z.); chinak_olga@niboch.nsc.ru (O.C.); kuprmax@niboch.nsc.ru (M.K.); golyshev@niboch.nsc.ru (V.G.); d.mitin@g.nsu.ru (D.M.); svetakh@niboch.nsc.ru (S.K.); 2Faculty of Natural Sciences, Novosibirsk State University, 630090 Novosibirsk, Russia

**Keywords:** triazinyl phosphoramidate, amphiphilic oligonucleotides, self-assembly, human serum albumin, Ku antigen, PARP1, protein–oligonucleotide complexes

## Abstract

Lipophilic oligonucleotide derivatives are a potent approach to the intracellular delivery of nucleic acids. The binding of these derivatives to serum albumin is a determinant of their fate in the body, as its structure contains several sites of high affinity for hydrophobic compounds. This study focuses on the features of self-association and non-covalent interactions with human serum albumin of novel self-penetrating oligonucleotide derivatives. The study revealed that the introduction of a triazinyl phosphoramidate modification bearing two dodecyl groups at the 3′ end region of the oligonucleotide sequence has a negligible effect on its affinity for the complementary sequence. Dynamic light scattering verified that the amphiphilic oligonucleotides under study can self-assemble into micelle-like particles ranging from 8 to 15 nm in size. The oligonucleotides with dodecyl groups form stable complexes with human serum albumin with a dissociation constant of approximately 10^−6^ M. The oligonucleotide micelles are simultaneously destroyed upon binding to albumin. Using an electrophoretic mobility shift assay and affinity modification, we examined the ability of DNA duplexes containing triazinyl phosphoramidate oligonucleotides to interact with Ku antigen and PARP1, as well as the mutual influence of PARP1 and albumin or Ku antigen and albumin upon interaction with DNA duplexes. These findings, together with the capability of dodecyl-containing derivatives to effectively penetrate different cells, such as HEK293 and T98G, indicate that the oligonucleotides under study can be considered as a platform for the development of therapeutic preparations with a target effect.

## 1. Introduction

One of the challenges of our era is to develop medicines that can ensure people’s longevity and an active lifestyle. Recent advancements in genetics and nucleic acid chemistry have enabled the development of fundamentally new therapeutic drugs and diagnostic tools that selectively affect molecular targets [1,2,3,4,5,6,7,8,9]. In oncotherapy, for example, short synthetic oligonucleotide duplexes are used as DNA repair inhibitors mimicking breaks to disorganize DSB repair pathways for sensitization of tumor cells to DNA-damaging therapies such as chemotherapy and radiotherapy [10,11]. There are two core pathways that repair double-strand breaks, the most harmful DNA damage, homologous recombination, and non-homologous end joining (NHEJ) [12,13,14]. NHEJ proceeds by classical (C-NHEJ) and alternative pathways (Alt-NHEJ). Ku is best known for its central role as a DNA end-binding factor at the initial stage of C-NHEJ, the main DNA double-strand break (DSB) repair pathway in mammalian cells. Alt-NHEJ is a minor pathway that requires PARP1 (poly(ADP-ribose)polymerase 1) as an indispensable protein; it operates when C-NHEJ for some reason is ineffective [14,15]. In addition, PARP1 is the key protein of base excision repair [16]. Both proteins efficiently bind DNA duplexes, displaying a high affinity for the double-strand ends [12,13,14], which indicates the importance of the development of novel duplex systems for a detailed investigation of interaction with those two proteins.

The attachment of hydrophobic residues to the oligonucleotide backbone, e.g., cholesterol, fatty acids, lipids, and alkyl-containing groups, is a promising approach to improving the pharmacokinetic properties of synthetic oligonucleotides [17,18,19,20,21,22]. The hydrophobic groups enhance the efficiency of membrane penetration of such derivatives into cells and improve their pharmacokinetic properties [23,24,25,26,27,28,29,30,31,32,33,34,35,36,37]. The enhanced pharmacokinetic characteristics and altered biodistribution are usually attributed to the binding of lipid-containing derivatives to serum proteins. Their association with serum proteins is believed to affect their cellular uptake and heavily influence their subsequent fate [25,37,38,39,40]. Among the variety of serum proteins, serum albumin (aSA), which contains at least seven fatty acid binding sites, is of particular interest [41]. SA is the major protein constituent of blood, accounting for 60% of the total protein content. This protein has gained the interest of researchers worldwide due to its low immunogenicity, long half-life, the ability to accumulate in tissues undergoing oncotransformation and inflammation, and the presence of binding sites for a variety of ligands in its structure [42,43,44,45,46]. Albumin significantly affects the pharmacokinetics and pharmacodynamics of drugs [47,48,49,50,51,52]. Furthermore, binding to this protein can increase the solubility of a chemical compound in the plasma or decrease its toxicity [53,54,55]. Due to the increased permeability and retention typical for the tumor, as well as the presence of specific receptors (Gp60, SPARC, FcRn) whose expression levels are increased in some cancers [45,46,56,57,58,59,60], albumin-binding constructs have great potential in cancer diagnosis and theranostics [42,44,61,62,63,64].

Oligonucleotides with lipophilic fragments introduced into the inter-nucleotide phosphate are an interesting type of derivative. Previously, dodecyl-containing NA structures were produced, which included phosphoramidate or non-nucleotide modifications with one to three dodecyl residues. Regardless of the modification methods employed, intracellular uptake levels were shown to increase with an increasing number of dodecyl residues [33]. Oligonucleotide derivatives containing three dodecyl residues in the non-nucleotide unit are also able to efficiently penetrate cells in vitro in the absence of transfection agents and form stable complexes with albumin under electrophoretic conditions [65,66]. This work reports on self-assembly and non-covalent interactions with human serum albumin (HSA) of amphiphilic triazinyl phosphoramidate oligonucleotide derivatives, functionalized with two dodecyl groups [67]. The derivatives were studied for their ability to form micelle-like structures and to associate with HSA. We investigated the interaction of DNA duplexes containing modified oligonucleotides with Ku antigen and PARP1, both in individual states and in whole-cell extracts. Additionally, the mutual influence of PARP1 and HSA or Ku antigen and HSA upon interaction with DNA duplexes was examined. Comprehensive in vitro experiments were conducted using dynamic light scattering (DLS), fluorescence spectroscopy, electrophoretic mobility shift assay (EMSA), affinity modification, and flow cytometry. The obtained results suggest a significant potential for the application of dodecyl-containing oligonucleotide derivatives for developing therapeutic drugs.

## 2. Materials and Methods

### 2.1. Materials

Acrylamide (AppliChem GmbH, Darmstadt, Germany); bis-acrylamide (Amresco, Solon, OH, USA); HSA (fraction V; Renal, Budapest, Hungary); Nile Red (9-diethylamino-5H-benzo[alpha]phenoxazine-5-one (C_20_H_18_N_2_O_2_)) (Merck, Darmstadt, Germany); 1 M MgCl_2_ (M1028-1ML) (Sigma, St. Louis, MO, USA); NaCl, (NH_4_)_2_S_2_O_8_ (Acros Organics, Waltham, MA, USA); N,N,N′,N′-tetramethyl ethylenediamine (TEMED) (Bio-Rad Laboratories, Berkeley, CA, USA); Tris (Fisher Scientific, Pittsburgh, PA, USA); AcOH (ice-cold) and Stains-All (Acros Organics, Waltham, MA, USA); xylene cyanol (Sigma, St. Louis, MO, USA); Ficoll 400 (Pharmacia, Stockholm, Sweden); DMEM; DMEM F12; Fetal Bovine Serum (FBS); GlutaMAX Supplement; Antibiotic-Antimycotic; and TrypLE (Gibco, Waltham, MA, USA) were used in this work. The isolation and purification of the HSA monomeric fraction were performed according to [68]. HSA modified by sulfo-Cy5 dye (HSA-Cy5) was obtained according to [69]. For the solutions, Milli-Q (18.2 M × Om/cm) water (purified by Simplicity 185 water system (Millipore, St. Louis, MO, USA)) was used.

### 2.2. Buffer Composition

PBS—10 mM Na_2_HPO_4_, 1.76 mM KH_2_PO_4_, 137 mM NaCl, 2.7 mM KCl, pH 7.4; TA—50 mM Tris-Acetate, pH 7.5; TAM—50 mM Tris-Acetate, pH 7.5, 15 mM MgCl_2_; TAN—50 mM Tris-Acetate, pH 7.5, 100 mM NaCl; Cacodylate Buffer—10 mM NaCac, pH 7.0, 100 mM NaCl; TE—10 mM Tris-HCl, pH 8.0, 1 mM EDTA, 50 mM NaCl; and TBE—25 mM Tris, 25 mM H_3_BO_3_, pH 8.3, 5 mM EDTA.

### 2.3. Oligonucleotide Synthesis

The phosphoramidite solid-phase synthesis of oligonucleotides was carried out on an ASM-800 synthesizer (Biosset, Novosibirsk, Russia). Oligonucleotides were synthesized at the 0.4 µmol scale, using standard commercial 2-cyanoethyl deoxynucleoside phosphoramidites and CPG solid supports (Glen Research, San Diego, CA, USA). Oligonucleotides without any fluorescent labels were cleaved from CPG with the mixture of aqueous ammonia and methylamine (1:1 (*v*/*v*), 55 °C, 0.5 h).

The insertion of triazinyl phosphoramidate modification bearing two dodecyl residues in appropriate oligonucleotide structures using the triazine modifier during the modified oxidation step was performed as described in [67]. The synthesis of the triazine modifier (2-azido-4,6-dichloro-1,3,5-triazine) was performed as described in [67].

For the introduction of FAM or carboxytetramethylrhodamine (TAMRA) residues, the corresponding phosphoramidite or modified CPG (Lumiprobe, Moscow, Russia) was used according to the manufacturer’s protocols. The CPG cleavage of FAM-containing oligonucleotides was first performed in aqueous ammonia (30% *m*/*v*, 15 min, 55 °C), and then aqueous methylamine (40% *m*/*v*, 15 min, 55 °C) was added equally to complete the process. The CPG cleavage of TAMRA-containing oligonucleotides was performed in the mixture of tert-butylamine/methanol/water (1:1:2, *v*/*v*/*v*) at 60 °C overnight.

### 2.4. Duplex Thermal Denaturation Experiments

Oligonucleotide duplexes were obtained by heating equimolar amounts of complementary strands in cacodylate buffer solution at 95 °C for 3 min, followed by incubation at 25 °C until complete cooling. Thermal denaturation experiments were performed using a Cary 300-BioMelt spectrophotometer equipped with a Peltier temperature-controlled cuvette holder (Varian, Sydney, Australia). Equimolar amounts (10 μM, 50 μL of each strand) of complementary oligonucleotides in cacodylate buffer were used. Melting curves were recorded at 260 and 270 nm over a temperature range of 5 °C to 95 °C at a heating rate of 0.5 °C/min.

### 2.5. Critical Aggregation Concentration (CAC) Determination by Nile Red Encapsulation Assay

Micelle formation of the modified oligonucleotide was characterized as described in the literature [65], using Nile Red as a fluorescent label. Modified or control oligonucleotides at 0.7 μM, 1 μM, 3 μM, 10 μM, 30 μM, and 50 μM concentrations were incubated with 100 μM Nile Red in TAM buffer at 25 °C for 3 h with stirring (350 rpm). After incubation, the samples were transferred to 96-well TPP^®^ plates (Sigma-Aldrich Chemie GmbH, Buchs, Switzerland). Nile Red fluorescence intensity spectra were obtained at room temperature using a CLARIOStar^®^ microplate reader (BMG LABTECH GmbH, Ortenberg, Germany). Fluorescence measurements were performed at an excitation wavelength of 550 nm and emission was monitored from 570 to 740 nm. The unmodified oligonucleotide was used as a control. The CAC was estimated by monitoring the fluorescence intensity of Nile Red versus the logarithm of the sample concentration. The CAC value was calculated by plotting the dependence of the Nile Red emission intensity at 630 nm on the log concentration (M) of oligonucleotides.

### 2.6. Characterization of Self-Assembled Micellar Structures by Dynamic Light Scattering (DLS)

The distribution of particles formed by dodecyl-containing oligonucleotides or their complexes with HSA by size was determined by DLS on a Zetasizer Nano-ZS (Malvern Panalytical Ltd., Malvern, UK) at 25 °C. Oligonucleotides (10 μM) as well as oligonucleotides with HSA in equimolar ratio were prepared in TAM buffer and size measurements were performed after incubation for 12 h.

### 2.7. Electrophoretic Mobility Shift Assay (EMSA)

Electrophoretic mobility shift assays were performed using a Thermo ScientificTM OwlTM Dual-Gel chamber (P8DS-2, Owl, Thermo Fisher Scientific Inc., Waltham, MA, USA) at 37 °C, 6 W for 3–4 h. Oligonucleotides were diluted to the desired concentration (10 μM) in 20 μL of the TA buffer solution and incubated for 1 h under stirring (450 rpm) before loading on a native polyacrylamide gel (PAAG) (8% PAAG, TAN). For albumin binding experiments, the protein was added to the oligonucleotides and incubated for 1 h before loading on the gel. After incubation, the reaction mixtures were supplemented with 2.5% Ficoll and loaded onto PAAG. Electrophoretic analyses of oligonucleotide duplexes and their complexes with HSA were performed in a similar manner. Oligonucleotide duplexes were formed by heating equimolar amounts of complementary strands in TAM buffer solution at 95 °C for 3 min, followed by incubation at 25 °C until complete cooling. After electrophoretic separation, the bands of oligonucleotides and HSA were visualized by Stains-All staining and in the case of FAM-labelled oligonucleotides, by scanning and image capture using a VersaDocTM MP 4000 Molecular Imager^®^ system (Bio-Rad, Berkeley, CA, USA) after excitation at 488 nm. For Stains-All staining (0.05% *w*/*v* Stains-All in 50% *w*/*v* formamide), the gel was stained in a dark chamber for 10 min after analysis. Decolorization was performed by removing the gel from the staining solution and exposing it to light until sufficient decolorization had occurred.

Electrophoretic analyses of 30-mer oligonucleotide duplexes with Ku antigen and PARP1 were performed using PAAG (8% PAAG (AA:BisAA = 40:1), 0.5X TBE) at 10 °C, 100 V. Protein–DNA binding was performed in the mixture of 50 mM Tris-HCl, pH 8.0, 20 mM NaCl, 0.25 mM EDTA, 5% glycerol and 0.025% NP-40 and incubated at 37 °C for 10 or 20 min as indicated in the figure legend. After incubation, the reaction mixtures were supplemented with 2.5% Ficoll and loaded onto PAAG. For quantitative analysis of autoradiography results, dried gels were exposed with a radiosensitive screen, and radioactive products were visualized and quantified using a Typhoon FLA 9000 scanner (GE Healthcare, Chicago, IL, USA) and Quantity One software v. 4.6.6 (Bio-Rad, Berkeley, CA, USA).

### 2.8. Determination of K_d_ by Fluorescence Spectroscopy

Oligonucleotides containing the FAM tag were diluted to the required concentration (10 µM, 70 µL) and protein was added in an excess of 0.1 to 3. Samples were incubated for 30 min at 25 °C and then transferred to a Costar-96 96-well plate. Fluorescence was measured at the endpoint at 530 nm after excitation at 483 nm on a Clariostar Monochromator Fluorimeter/Luminometer/Spectrophotometer (BMG LABTECH, Germany). The K_d_ value was calculated by plotting the dependence of the FAM emission intensity at 530 nm on the concentration (µM) of HSA. The resulting dependence was approximated by the following function:Y = F_max_ × X/(K_d_ + X),(1)
where F_max_ is the maximum fluorescence intensity.

### 2.9. 5′-dRp-DNA Obtaining

Annealing of DNA duplexes, introduction [^32^P]-labels at the 5′ end of oligonucleotides, and treatment with UDG to obtain 5′-dRp-DNA were carried out according to [70]. In brief, 5′-[^32^P]-phosphate residues were introduced using T4 polynucleotide kinase. Phosphorylated oligonucleotides were purified by 20% PAAG-7M urea electrophoresis [71]. To obtain DNA duplexes, the corresponding oligonucleotides were mixed in an equimolar ratio in a buffer containing 10 mM Tris-HCl (pH 8.0), 1 mM EDTA, and 50 mM NaCl; the reaction mixture was heated for 5 min at 90 °C, and then slowly cooled down to room temperature. UDG treatment of DNA containing dUMP was carried out immediately prior to an experiment: 1 pmol of DNA was incubated with 0.1 U UDG for 30 min at 37 °C.

### 2.10. Affinity Modification of Ku Antigen, PARP1, and DNA Polymerase β by 5′-dRp DNAs

Ku antigen, PARP1, and DNA polymerase β were purified as described in [72,73,74], respectively. Whole-cell extracts (WCEs) were prepared according to [75]. Modification of proteins by 5′-dRp DNAs and following analysis of the cross-linking products were performed as in [70,76]. Briefly, for DNA-protein cross-linking, the proteins at the concentrations indicated in the figure legends were incubated with 100 nM dRp-containing DNAs in the following buffer: 50 mM Tris-HCl, pH 8.0, 20 mM NaCl, 10 mM EDTA, 5% glycerol and 0.025% NP-40, for 10 min at 37 °C. Then the reaction mixtures were supplemented with 20 mM NaBH_4_ following incubation at 0 °C for 30 min. The mixtures were supplemented with Laemmli sample loading buffer, heated for 5 min at 95 °C, and loaded onto PAAG.

### 2.11. Cell Lines

HEK293 (human embryonic kidney 293 cells) and T98G (glioblastoma cells) were cultured in culture medium (DMEM (HEK293 cells) or DMEM F12 (T98G cells)) supplemented with 10% FBS, GlutaMAX Supplement (200 mM L-alanyl-L-glutamine) and antibiotic/antimycotic mixture (100 units/mL penicillin, 0.1 mg/mL streptomycin and 0.25 μg/mL amphotericin) at 37 °C in 5% CO_2_.

### 2.12. Cellular Accumulation Assay

One day before transfection, cells were seeded at 1.2 × 10^5^ cells per well in a culture medium in a 24-well plate. Cells were washed with PBS, then PBS was replaced with 250 μL/well of fresh DMEM containing one of five compositions: FAM-TZ16/M16; TZ30/M30-FAM; HSA-Cy5; HSA-Cy5 + FAM-TZ16/M16; or HSA-Cy5 + TZ30/M30-FAM with final concentration of HSA and duplexes of 5 μM. Oligonucleotide duplexes were formed by heating equimolar amounts of complementary strands in TA buffer solution at 95 °C for 3 min, followed by incubation at 25 °C until complete cooling. After 4 h, cells were washed and detached from the plate using TrypLE, resuspended in DMEM supplemented with 10% FBS, and then cells were centrifuged at 800 rpm for 5 min. Cells were analyzed on a BD FACSCanto II flow cytometer (BioLine, Saint Petersburg, Russia).

## 3. Results and Discussion

### 3.1. Oligonucleotides and Their Derivatives Used in This Study

In the initial phase, we synthesized oligonucleotide derivatives of 16-, 17-, and 30-mers, each carrying the triazinyl phosphoramidite (TZD) modification with two dodecyl residues (Figure 1A,B, oligonucleotides FAM-TZD16, FAM-TZD17 and TZD30; all of the other oligonucleotide structures were used as the controls for the experiment). Results of mass-spectrometry of modified oligonucleotides TZ30, FAM-TZ16 and FAM-TZ17 are shown in Appendix A. The introduction of such a modification was carried out during standard solid-phase synthesis, with only one stage change [67]. That allowed us to simplify the scheme of the synthesis of modified oligonucleotides, avoiding the post-synthetic introduction of the modification. The inter-nucleotide phosphate group differs favorably from other positions for the introduction of non-natural chemical groups, not only since there are numerous approaches that allow the obtaining of NA-derivatives with high efficiency [77,78,79,80,81,82], but also because modifications of the phosphate backbone insignificantly affect the ability of oligonucleotides to form complementary complexes with target nucleic acids. The 16-mer of triazinyl phosphoramidate oligonucleotide, which also contains fluorescein (FAM) moiety in its structure, has already been studied for its efficiency of intracellular accumulation [67]. Considering the high level of demonstrated cellular delivery, we have decided to investigate these derivatives in terms of self-assembly into micelle-like associates, non-covalent binding to HSA, and the ability of DNA duplexes containing TZD-modified oligonucleotides to interact with Ku antigen and PARP1, even though the FAM-moiety could potentially influence these properties.

### 3.2. Hybridization Properties of Modified Oligonucleotides

The melting temperature (T_m_) of the complementary complexes under conditions closely resembling physiological ones demonstrated negligible differences in thermal stability, with T_m_ being slightly decreased for the duplexes FAM-TZD16/M16 and FAM-TZD17/TAMRA-M27 and increased for TZD30/M30-FAM with modified oligonucleotides (Table 1, Appendix A). Therefore, the presence of triazinyl phosphoramidate with two dodecyl groups in the sugar-phosphate backbone appears to have little impact on the oligonucleotide’s affinity for the complementary sequence. It should be noted that the derivative melting curves of the FAM-TZD16/M-16 sample show an additional peak (Appendix A) that indicates the presence of stable intramolecular secondary structures.

### 3.3. CAC Determination by Nile Red Encapsulation Assay

The amphiphilic nature of the oligonucleotide derivatives under study suggests their tendency to form micelle-like structures. This was confirmed by further testing with the use of the Nile Red encapsulation assay to determine their critical aggregation concentration (CAC). In most polar solvents, the Nile Red dye does not fluoresce. However, it shows intense fluorescence in lipid-rich media [83]. Incubation of a defined amount of dye with increasing concentrations of FAM-TZD16, FAM-TZD17, and TZD30 resulted in an amplification of the fluorescence intensity of Nile Red (Appendix A) that indicates its encapsulation in the nonpolar microenvironment of the micellar structure. Conversely, the control oligonucleotide without dodecyl chains displayed no notable increase in the fluorescence intensity of Nile Red (Appendix A). The CAC of micellar particles was found to be 8.1 ± 0.3 μM and 8.9 ± 0.9 μM for FAM-TZD16 and FAM-TZD17, correspondingly. Whereas the CAC reduced slightly to 7.0 ± 1.0 μM upon the increase of oligonucleotide length to 30 monomeric units.

### 3.4. The Size Assessment of Oligonucleotide Micelle-like Structures and Their Complexes with HSA by DLS

As a result of their amphiphilic properties, the oligonucleotides under study are capable of self-assembling into micelle-like particles, which in aqueous solutions consist of a hydrophobic dodecyl core surrounded by a hydrophilic oligonucleotide corona. To stabilize the outer hydrophilic layer, positively charged ions are utilized to lower electrostatic repulsion between negatively charged phosphate groups. The size of the micelle-like structures was assessed using DLS measurements (Table 2). After being pre-incubated in TAM buffer, two types of particles were detected in the FAM-TZD17 sample having an average hydrodynamic diameter (D_h_) of 8.6 ± 2.5 nm and 11.6 ± 3.1 nm, in contrast to the 0.71 ± 0.13 nm of the 17-mer control oligonucleotide lacking the triazinyl phosphoramidate modification with dodecyl chains (Appendix A). Similar results were obtained for the 30-mer oligonucleotide. TZD30 particles with hydrodynamic diameter values of 10.4 ± 3.0 nm and 14.1 ± 4.0 nm were observed (Appendix A). At the same time, the average size of FAM-TZD16 particles was 11.9 ± 3.4 nm (Appendix A).

The existence of several particle populations in the samples of FAM-TZD17 and FAM-TZD30 under study may be related to an altered conformation of the oligonucleotide, which is caused by intramolecular pairing. The OligoAnalyzer software v. 1.0.2 was used to predict the possible secondary structures formed by each of the sequences under the experimental conditions (TAM buffer, 10 µM oligomer concentration) (Appendix A). As can be seen, the most stable among these is the hairpin that is formed by the 16-mer oligonucleotide (Appendix A). To confirm this hypothesis, a particle size measurement for TZD30 was carried out at 40 °C, the temperature that is above the melting point of the hairpin in the oligonucleotide. Indeed, under these conditions, only one population of particles with a hydrodynamic diameter of 14.8 ± 4.0 nm was observed, indicating the absence of particles formed by oligonucleotides in conformation with the secondary structure. This confirmed the hypothesis. At the same time, for 17-mer oligonucleotides, a similar picture would most likely be observed, but the melting temperature of the hairpin formed by itself exceeds the normal temperature of the human body. Therefore, the verification has no practical significance.

In the literature, there is a focus on the correlation between the size of micelle-like structures and the length and nucleotide sequence of the oligonucleotide [34,65,84]. Our research revealed that the size of self-assembled oligonucleotide particles is dependent on their nucleotide composition and, in particular, on their ability to form intramolecular secondary structures. Specifically, when less thermodynamically stable hairpins were formed by the 17- and 30-mer oligonucleotides, two types of particles were observed. The formation of a more thermodynamically stable hairpin by the 16-mer oligonucleotide led to the existence of only one particle population. The size of the micelles formed by the oligonucleotides under study ranged from 8 to 15 nm and correlated with the length of the oligonucleotide sequence. The average hydrodynamic diameter values obtained are typical of those previously reported for nucleic acid micelles in aqueous solution [34,85,86,87]. It has been previously reported that bovine serum albumin (BSA) has a disruptive effect on nucleic acid micelles [88]. Indeed, we observed the same effect for HSA. The average hydrodynamic diameter of HSA–oligonucleotide complexes decreased (Table 2, Appendix A). It is worth noting that the size of the protein associated with the oligonucleotides under study was found to be 6.0 ± 1.3 nm, which is close to the free HSA.

### 3.5. Detection of HSA–Oligonucleotide Complexes by EMSA

Human serum albumin is a well-known carrier of both endogenous and exogenous compounds [41]. Its structure enables it to bind easily to compounds of diverse nature, including nucleic acids. The literature data suggests that the presence of modifications in the sugar-phosphate backbone of oligonucleotides significantly improves their interaction with HSA [89,90]. Moreover, lipophilic derivatives with high affinity for albumin are characterized by improved pharmacokinetic properties, demonstrating higher efficacy in comparison to unmodified analogs [25,38,39,40,90,91,92]. Complexes of HSA with the oligonucleotide derivatives under study were investigated by EMSA (Figure 2, Figure 3 and Figure 4). FAM-TZD17 forms complexes with HSA that are stable under the electrophoretic conditions used (Figure 2, lanes 3–6). The resulting associates have a stoichiometry of roughly 1:1 (Figure 2, lane 4). As anticipated, the duplex also binds to HSA (Figure 2, lanes 9–12), but with lower efficiency. When the complementary chain TAMRA-M27 was added to the preincubated mixture of HSA with FAM-TZD17, the protein was displaced from the complex, as the bands corresponding to the duplex in the zone of its characteristic mobility were observed (Figure 2, lanes 9–12).

A similar experiment was carried out to study the interaction of HSA with FAM-TZD16. It is difficult to estimate the possible stoichiometry of binding in this case because the FAM-TZD16 has lower mobility than HSA (Figure 3, lanes 1 and 2), which is associated with the formation of a stable secondary structure. However, we observed that when FAM-TZD16 binds to HSA, the complex exhibits the same mobility as the protein (Figure 3, lanes 3–6). Regarding the duplex, its partial melting was observed under the electrophoretic conditions used, as the bands of complementary sequence in the zone of its characteristic mobility were detected (Figure 3, lanes 9–12). Nonetheless, the FAM-TZD16 remains in the complex with the protein with almost the same mobility as the protein (Figure 3, lanes 9–12).

Similar patterns were observed for the interaction of TZD30 with HSA (Figure 4, lanes 5–8). As the duplex is highly thermostable, it not only avoids melting but also completely binds to the protein at a ratio of 1:1 (Figure 4, lane 6). To summarize, the binding of duplex structures to albumin relies heavily on their stability. Duplexes with shorter sequences melt more easily, causing the complementary chain to displace. However, an increase in sequence length raises the stability of the corresponding duplex, leading to its complete binding to HSA.
Figure 4Detection of HSA complexes with TZD30/M30-FAM by EMSA; oligonucleotide samples contained 10 µM oligomer; * indicates an ON:HSA ratio. HSA (Lane 1), M30-FAM (Lane 2), TZD30 (Lane 3), TZD30/M30-FAM (Lane 4), and TZD30/M30-FAM + HSA (Lanes 4–8). Region-corresponding mobilities of the HSA–oligonucleotide complexes under the conditions used are marked with a blue curly bracket, and the range of mobility of free oligonucleotides is marked with a red curly bracket.
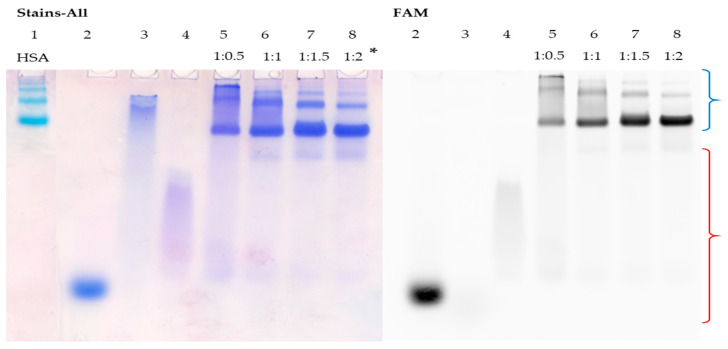


### 3.6. K_d_ Determination of HSA–Oligonucleotide Complexes by Fluorescence Titration Experiment

The dissociation constant of the HSA–oligonucleotide complexes with the triazinyl phosphoramidate modification under study was estimated by monitoring the change in fluorescein fluorescence intensity, which increased with increasing HSA concentration (Appendix A). The dissociation constants obtained were 1.2 ± 0.2 μM and 1.5 ± 0.1 μM for the complexes FAM-TZD17/HSA and FAM-TZD16/HSA, respectively. It is worth noting that increased affinity for serum proteins can not only have a positive effect on the biodistribution of modified derivatives and their penetration into cells, but can also significantly reduce the biological activity of the therapeutic oligonucleotides [39]. According to Chappell et al., an increase in serum protein concentration in the culture medium had a negligible impact on the activity of native antisense oligonucleotides. However, it significantly reduced the activity of oligonucleotides modified with a palmitic acid residue. The authors demonstrated that the enhanced affinity for albumin resulted in the inhibition of the release of the derivatives under study in the interstitium and their excretion from the tissues with albumin.

### 3.7. Affinity Modification of Ku Antigen, PARP1 and DNA Polymerase β by [^32^P]5′-dRp-DNAs

Affinity modification, a very promising approach in the search for cellular proteins interacting with DNA duplexes under investigation, consists of using their chemically reactive derivatives of DNA, which can form covalent complexes of DNA-protein thereby marking the interacting protein [76,93].

Deoxyribose in the AP site and the 5′-dRp (5′-deoxyribosephosphate) group exist in an equilibrium of cyclic furanose and acyclic aldehyde forms; the last is capable of reacting with a primary amino group of protein in its vicinity. Forming the Schiff base intermediate is characteristic of the catalytic action of AP/5′-dRp lyases. The Schiff base intermediate can be reduced by NaBH_4_, which leads to the formation of an irreversible protein/DNA complex between the enzyme and DNA. Both Ku antigen and PARP1 are known to have 5′-dRp lyase activity, thus DNAs containing the 5′-dRp group can be used as chemically reactive DNA probes for marking these proteins in cell extracts [70,76,93,94].

First, using a DNA duplex composed of two radioactively labeled 30-mer oligonucleotides bearing 5′-dRp in both chains, we modified the proteins of WCEs of HEK293T and HEK293A cells and purified the putative protein targets (Appendix A). As expected, Ku antigen, PARP1, and DNA polymerase β are the main protein targets of the 5′-dRp-DNA duplex in the extracts. Effective labeling of predominantly these three proteins in the extracts appears to be due to the high affinity of the proteins for a specific DNA substrate and their high copy number in cells. Indeed, these proteins have been demonstrated to be rather abundant [95]. It should be noted that some proteins, not known as lyases, are able to interact with the 5′-dRp residue forming the Schiff base [93].

As can be seen, for each of the purified proteins, more than one product of DNA-crosslinking to the protein was observed. There are several reasons for the appearance of multiple products. It should be noted that the difference in electrophoretic mobility of the products due to oligonucleotide excision is greater for proteins with lower molecular weight. Illustration and explanation of the product formation are represented in Appendix A.

For Pol β, lower electrophoretic mobility product appears to correspond to a polypeptide with a covalently attached oligonucleotide while faster-moving abundant product corresponds to a polypeptide with an attached sugar residue, the product arising after β-elimination has occurred. We have earlier demonstrated Pol β and apurinic/apyrimidinic endonuclease 1 existence of such products [96]. Kinetic analysis revealed that the formation of the Schiff-base intermediate and β-elimination occur much faster than hydrolysis of the Schiff-base that restores the initial form of the enzyme. In addition, during sample preparation for analysis by the Laemmli method [97], no special compounds to disrupt hydrogen bonds between DNA bases are added to the sample, therefore, complete melting of the duplex does not always take place. The attachment of the oligonucleotide to different lysines within one polypeptide chain may also interfere with product mobility.

For Ku antigen, two products corresponding to Ku70 and Ku80 subunits cross-linked with oligonucleotide were observed. As earlier shown, the lysine(s) in the Ku70 N-terminal part are mainly involved in realization of the AP/dRp lyase activity [94]. In addition, it has been demonstrated that the lysine(s) of Ku80 can substitute for the mutated lysines of Ku70 in AP site cleavage with moderate drop of activity [94,98]. Lysine 31 of Ku70 appears to be the main nucleophile involved in the catalytic action. However the additional mutation of K160 in Ku70 and six other lysines in Ku80 is necessary for complete y elimination of the activity [99].

### 3.8. Studying the Mutual Influence of PARP1 and HSA or Ku Antigen and HSA upon Interaction with DNA Duplexes by EMSA

To study the mutual influence of PARP1 and HSA or Ku antigen and HSA upon binding with DNA duplexes, we used EMSA, but conditions were optimized for the study of proteins having high affinity for DNA. Using the radioactively labeled DNA allows for the detection of complexes at concentrations lower than the micromolar level, thus minimizing unspecific binding detection. Considering the relatively low affinity of HSA to DNA (Kd in micromolar level) and the absence of HSA influence on the interaction of PARP1 with DNA at low micromolar level (Appendix A), in the study we used 15 μM HSA.

Using EMSA for the studies of PARP1 and Ku antigen binding with DNA has a number of features. The binding of Ku antigen with DNA duplexes depends on their length. The minimal length required for Ku binding is 14 base pairs and two Ku molecules can be bound with the duplexes of about 30 base pairs [100]. Typical examples of Ku binding with DNA duplexes are shown in Figure 5. Unfortunately, the bands of the oligonucleotide complexes of HSA and Ku with TZD-modified oligonucleotide do not resolve enough in the gel. It should be noted that under used conditions of EMSA, no complexes of HSA with regular DNA duplex were observed, while for TZD-modified DNA such complexes are detected, but smearing of the bands may reflect the dissociation of these complexes during electrophoresis. Both for the non-modified 30-mer DNA duplex and for the TZD-modified one, simultaneous addition of 15 μM HSA and 30 or 60 nM Ku does not reduce Ku binding. Preincubation of DNAs with 15 μM HSA for 15 min followed by the addition of Ku also does not impair Ku binding.

Data on the cross-linking of [^32^P]-dRp-containing DNAs to Ku antigen and HSA are shown in Figure 6. Unexpectedly, HSA is able to form the Schiff-base intermediate with the dRp-group (Figure 6, lanes 5 and 9). The yield of products is higher for TZD-containing DNA, indirectly testifying to more efficient binding of this DNA to HSA. Preincubation of dRp-DNA with 15 μM HSA under equilibrium conditions appears to result in the cross-linking of a considerable proportion of TZD-containing DNA to HSA, thereby decreasing the yield of Ku-DNA cross-linking products. dRp-DNA without TZD-modification forming fewer cross-linking products with HSA keeps the capability to interact with Ku after its addition. Ku antigen and HSA, when added simultaneously, compete for DNA which results in a decrease of DNA-protein cross-linking products both for HSA and Ku (Figure 6, compare lanes 2, 3, and 5) with the effect being more pronounced for HSA.

Data demonstrating the effect of HSA on DNA binding to PARP1 are presented in Figure 7. It should be noted that the complexes of PARP1 with duplexes of about 30 base pairs have very low mobility in the gels, and the complexes containing more than one PARP1 molecule could not be distinguished from the 1:1 ratio. Data demonstrate that TZD-containing DNA more efficiently binds with proteins than regular DNA. No complexes of HSA with regular DNA were detected (Figure 7, lanes 6 on both panels). At 60 nM PARP1, practically no free TZD-containing DNA when PARP1 or PARP1 + HSA were present, while free regular DNA was observed (Figure 7, left panel: compare lanes 2–5 with lanes 6–9). When the concentration of PARP1 is higher than that of DNAs (150 nM and 100 nM, respectively, right panel) in all cases practically no free DNAs were observed if PARP1 was present. In all cases, the presence of HSA does not lower the amount of PARP1 bound to DNA.

Data on DNA cross-linking to PARP1 and HSA are shown in Figure 8. The yield of products is higher for TZD-containing DNA, indirectly testifying to more efficient binding of this DNA to HSA. When TZD-containing DNA was first preincubated with HSA and then PARP1 was added followed by incubation, both products of PARP1 and HSA cross-linking were observed (Figure 8, lanes 4 and 8), with the yield of PARP1 cross-linking being lower (Figure 8, compare lanes 2 and 4).

The possibility of penetration into cells of DNA duplexes bound within complexes with HSA raises a question about the accessibility of these DNAs for target cellular proteins. Thus, the formation of HSA complexes with DNA duplexes apparently would not interfere with the complex formation of DNA with Ku. For PARP1, the tendency resembles that observed with Ku.

### 3.9. Intracellular Accumulation of DNA Duplexes Bound within Complexes with HSA

Previously, the ability of an oligonucleotide containing two dodecyl residues to enter cells in the absence of transfectants was shown [67]. Low cytotoxicity of the studied lipophilic oligonucleotide derivatives was shown ([67] and Appendix A). To investigate the ability of dodecyl-modified oligonucleotides TZD16 and TZD30 in complexes with HSA to penetrate cells, we examined the accumulation of FAM-TZD16/M16 + HSA-Cy5 and TZD30/M30-FAM + HSA-Cy5 complexes in HEK293 and T98G cells using flow cytometry. These cell lines were chosen due to significant differences in their albumin receptor expression, according to the database at www.proteinatlas.org (accessed on 7 November 2023). For example, T98G cells have higher levels of SPARC compared with HEK293 cells.

Both TZD16 and TZD30 oligonucleotides effectively penetrated HEK293 and T98G cells on their own (Figure 9). When combined with HSA, oligonucleotides also penetrated cells effectively. Although the presence of HSA had different effects on TZD16 and TZD30 in T98G cells, the distribution of fluorescent cells treated with TZD30/M30-FAM and its complex with HSA looked similar (Figure 9D), while cells incubated with complex FAM-TZD16/M16 + HSA-Cy5 had higher fluorescent intensity compared with FAM-TZD16/M16 (Figure 9C). Thus, we can assume that complex FAM-TZD16/M16 with HSA may have different penetration mechanisms in the investigated cell lines.

The fluorescence intensity comparison did not reveal a significant difference in cell penetration efficiency between FAM-TZD16/M16 + HSA-Cy5 and TZD30/M30-FAM + HSA-Cy5 (Appendix A). However, it was observed that the penetration of HSA-Cy5 in HEK293 cells was equal to the penetration of complexes, while in T98G cells, the penetration of HSA-Cy5 was stronger than both complexes. Thus, oligonucleotides significantly decrease the penetration of HSA in glioblastoma cells. This suggests that the penetration of HSA–oligonucleotide complexes may not solely be provided by human serum albumin receptors. Thus, both TZD16 and TZD30 oligonucleotides, whether alone or in complexes with HSA, are capable of effectively penetrating different cells, such as HEK293 and T98G.

## 4. Conclusions

To summarize, in this study, we analyzed the key features of self-assembly into micelle-like structures of novel oligonucleotide derivatives containing triazinyl phosphoramidate substituted by two dodecyl groups. The modified oligonucleotides efficiently bind to human serum albumin under the conditions used. DNA duplexes containing TZD-modified oligonucleotides are capable of efficiently interacting with Ku antigen and PARP1, both in individual states and in whole-cell extracts. HSA, even at micromolar concentrations, does not compete with Ku antigen and PARP1 for DNA. These results, together with the capability of triazinyl phosphoramidate oligonucleotides to effectively penetrate different cells and the already known facts that they have increased resistance to cellular nucleases and do not exhibit significant cytotoxicity [67,101], make them very attractive candidates for the development of highly selective therapeutic agents.

## Figures and Tables

**Figure 1 pharmaceutics-15-02779-f001:**
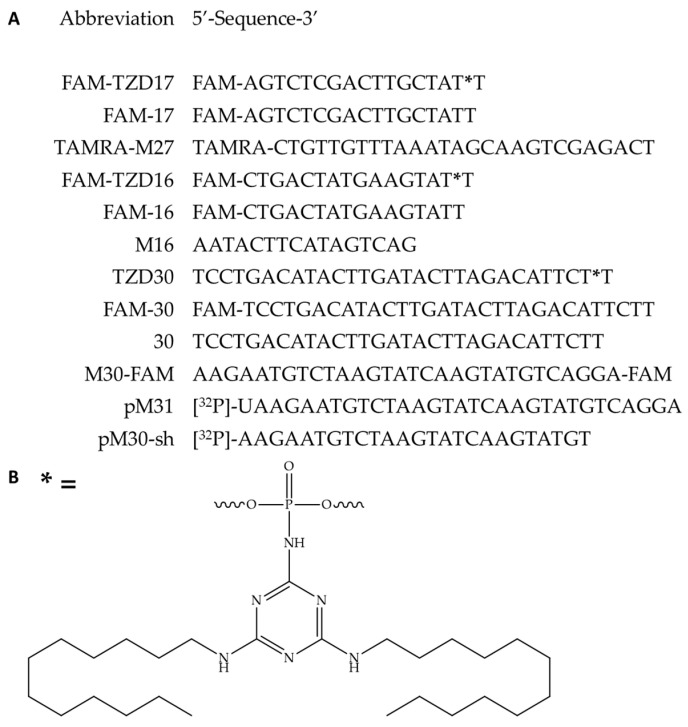
The oligonucleotide derivatives’ sequences (**A**), the structure of the triazinyl phosphoramidate modification (**B**); * indicates a position of the triazinyl phosphoramidate modification (**B**). All oligonucleotides are deoxy. Here, FAM represents the 6-carboxyfluorescein residue and TAMRA—carboxytetramethylrhodamine residue.

**Figure 2 pharmaceutics-15-02779-f002:**
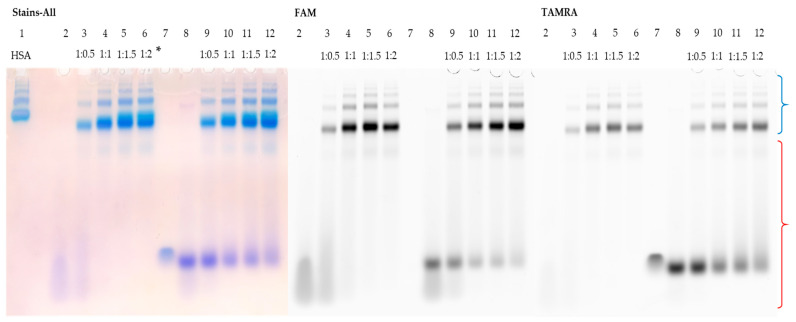
Detection of HSA complexes with FAM-TZD17 and FAM-TZD17/TAMRA-M27 by EMSA; oligonucleotide samples contained 10 µM oligomer; * indicates an ON:HSA ratio. HSA (lane 1), FAM-TZD17 (lane 2), FAM-TZD17 + HSA (lanes 3–6), TAMRA-M27 (lane 7), FAM-TZD17/TAMRA-M27 (lane 8), and FAM-TZD17/TAMRA-M27 + HSA (lanes 9–12). Region-corresponding mobilities of the HSA–oligonucleotide complexes under the conditions used are marked with a blue curly bracket, and the range of mobility of free oligonucleotides is marked with a red curly bracket.

**Figure 3 pharmaceutics-15-02779-f003:**
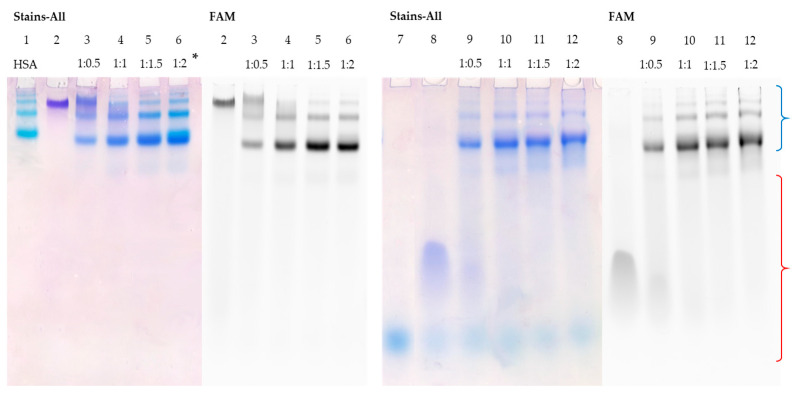
Detection of HSA complexes with FAM-TZD16 and FAM-TZD16/M16 by EMSA; oligonucleotide samples contained 10 µM oligomer; * indicates an ON:HSA ratio. HSA (lane 1), FAM-TZD16 (lane 2), FAM-TZD16 + HSA (lanes 3–6), M16 (lane 7), FAM-TZD16/M16 (lane 8), and FAM-TZD16/M16 + HSA (lanes 9–12). Region-corresponding mobilities of the HSA–oligonucleotide complexes under the conditions used are marked with a blue curly bracket, and the range of mobility of free oligonucleotides is marked with a red curly bracket.

**Figure 5 pharmaceutics-15-02779-f005:**
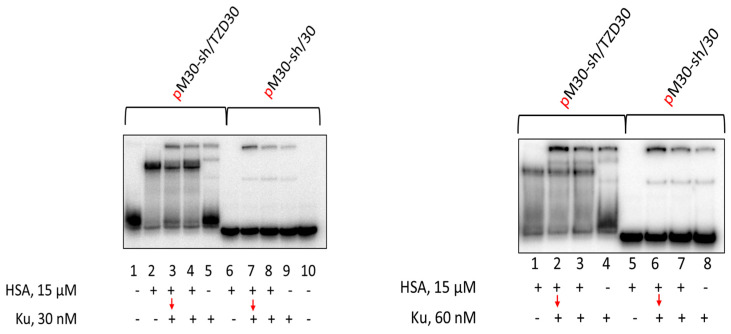
EMSA of Ku-DNA complexes. 100 nM DNAs were incubated at 37 °C for 20 min with 15 μM HSA ((**left**) panel: lanes 2, 6; (**right**) panel: lanes 1, 5) or for 10 min with 15 μM HSA + 30/60 nM Ku ((**left**) panel: lanes 4, 8; (**right**) panel: lanes 3, 7), or 10 min with 15 μM HSA ((**left**) panel: lanes 3, 7; (**right**) panel: lanes 2, 6) followed by addition of 30/60 nM Ku with further incubation for 10 min, or 20 min with 30/60 nM Ku ((**left**) panel: lanes 5, 9; (**right**) panel: lanes 4, 8). Lanes 1 and 10 ((**left**) panel)—DNA, control without protein(s).

**Figure 6 pharmaceutics-15-02779-f006:**
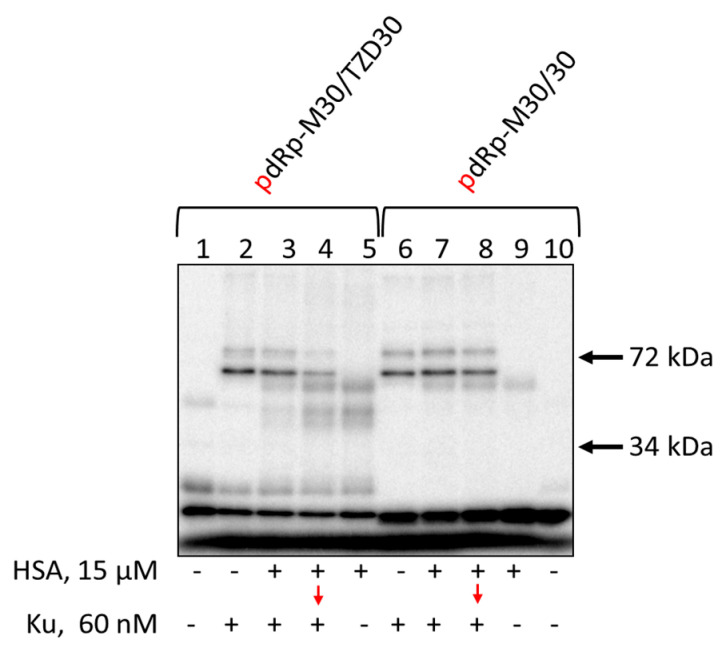
Cross-linking of [^32^P]-dRp-containing DNAs to Ku and HSA. 100 nM [^32^P]-dRp-containing DNAs were incubated at 37 °C for 20 min with 15 μM HSA (lanes 5, 9) or for 10 min with 15 μM HSA + 60 nM Ku (lanes 3, 7), or for 10 min with 15 μM HSA (lanes 4, 8) followed by addition of 60 nM Ku with further incubation for 10 min or 20 min with 60 nM Ku ARP1 (lanes 2, 6). Lanes 1 and 10—DNA, control without protein(s). After incubation, the reaction mixtures were supplemented with 20 μM sodium borohydride to reduce the Schiff base for 30 min at 0 °C.

**Figure 7 pharmaceutics-15-02779-f007:**
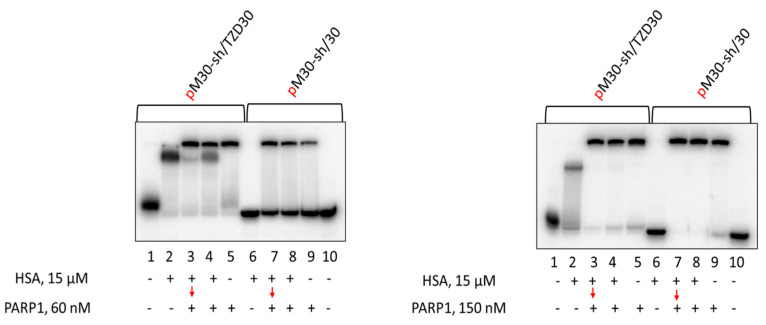
EMSA analysis of PARP1-DNA complexes. 100 nM DNAs were incubated at 37 °C for 20 min with 15 μM HSA (lanes 2, 6) or 10 min with 15 μM HSA + 60/150 nM PARP1 (lanes 4, 8), or 10 min with 15 μM HSA (lanes 3, 7) followed by addition of 60/150 nM PARP1 with further incubation for 10 min or 20 min with 60/150 nM PARP1 (lanes 5, 9). Lanes 1 and 10—DNA, without protein(s).

**Figure 8 pharmaceutics-15-02779-f008:**
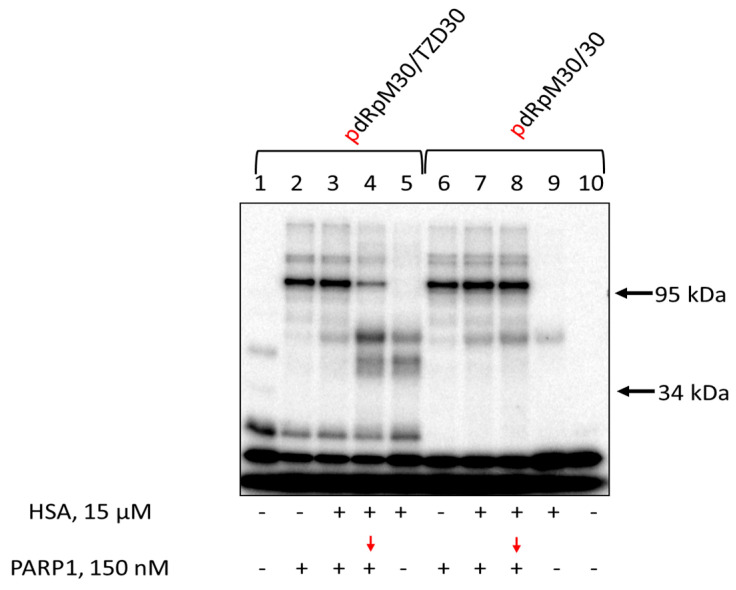
Cross-linking of [^32^P]-dRp-containing DNAs to PARP1 and HSA. 100 nM [^32^P]-dRp-containing DNAs were incubated at 37 °C for 20 min with 15 μM HSA (lanes 5, 9) or 10 min with 15 μM HSA + 0.15 μM PARP1 (lanes 3, 7), or 10 min with 15 μM HSA (lanes 4, 8) followed by addition of 0.15 nM PARP1 with further incubation for 10 min or 20 min with 0.15 μM PARP1 (lanes 2, 6). Lanes 1 and 10—DNA, control without protein(s). After incubation, the reaction mixtures were supplemented with 20 mM sodium borohydride to reduce the Schiff base for 30 min at 0 °C. The products were analyzed in 10% SDS-PAGE according to [97].

**Figure 9 pharmaceutics-15-02779-f009:**
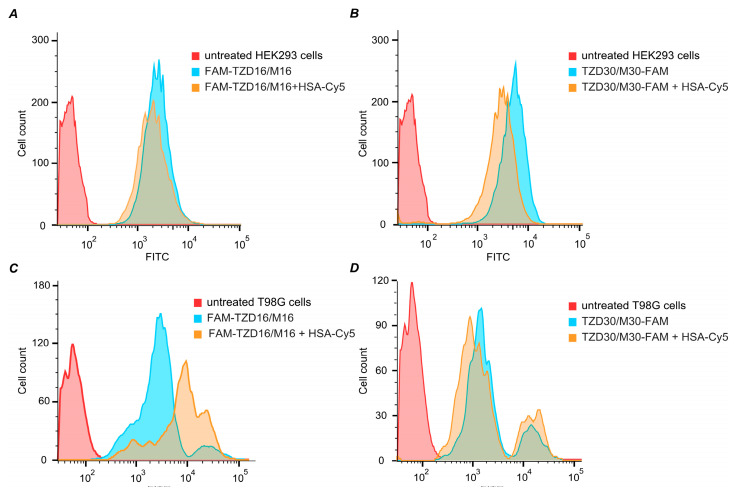
Fluorescence intensity of HEK293 (**A**,**B**) and T98G (**C**,**D**) cells treated with 5 µM of oligomers TZD16 or TZD30 alone and within complexes with HSA for 4 h.

**Table 1 pharmaceutics-15-02779-t001:** Thermodynamic characteristics of duplex structures under study.

Sample	Parameter	
	T_m_, °C	ΔG^0^, kcal × mol^−1^
FAM-16/M-16	51.9 ± 0.1	(−1.34 ± 0.01) × 10^4^
FAM-TZD16/M-16	48.7 ± 0.3	(−1.15 ± 0.03) × 10^4^
FAM-17/TAMRA-M27	60.9 ± 0.2	(−1.89 ± 0.09) × 10^4^
FAM-TZD17/TAMRA-M27	59.9 ± 0.1	(−1.57 ± 0.03) × 10^4^
FAM-30/M30-FAM	64.6 ± 0.5	(−1.91 ± 0.05) × 10^4^
TZD30/M30-FAM	65.3 ± 0.5	(−2.1 ± 0.2) × 10^4^

**Table 2 pharmaceutics-15-02779-t002:** The average hydrodynamic diameter (D_h_) of the micellar particles.

Sample	D_h_, nm	PDI
FAM-17	0.71 ± 0.13	0.249
FAM-TZD16	11.9 ± 3.4	0.303
FAM-TZD17	8.6 ± 2.5	0.265
	11.6 ± 3.1	0.531
TZD30	10.4 ± 3.0	0.190
	14.1 ± 4.0	0.245
TZD30 (T = 40 °C)	14.8 ± 4.0	0.195
HSA	6.0 ± 1.3	0.284
HSA/FAM-TZD16	6.7 ± 2.0	0.288
HSA/FAM-TZD17	7.4 ± 2.0	0.423
HSA/TZD30	7.0 ± 2.1	0.226

## Data Availability

Data are contained within the article and Appendix A.

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
