# Peer review of "Self-Penetrating Oligonucleotide Derivatives: Features of Self-Assembly and Interactions with Serum and Intracellular Proteins"

_pharmaceutics, 2023, doi:10.3390/pharmaceutics15122779_

Round 1

Reviewer 1 Report

Comments and Suggestions for Authors

The manuscript “Self-penetrating oligonucleotide derivatives: features of self-assembly and interactions with serum and intracellular proteins” by Dmitrienko et al, related to lipophilic oligonucleotide derivatives as a potent approach to the intracellular delivery  of nucleic acids. DNA duplexes containing amphiphilic triazinyl phosphoramidate oligonucleotide derivatives, functionalized with two dodecyl were investigated in terms of self-assembly into micelle-like associates, non-covalent binding to HSA, and the ability of DNA duplexes containing TZD-modified oligonucleotides to interact with Ku antigen and PARP1.

Overall, the work is well structured and presented clearly to the reader, the article is recommended for publication with minor revisions.

The introduction is well developed and consistent with the topic of the manuscript, nevertheless the beneficial conjugation of lipophilic moieties as bile acids to oligonucleotides, should be considered  (Int. J. Mol. Sci. 2022, 23, 4270. https://doi.org/10.3390/ijms23084270)

Line 255: the expression rationalize should be substituted with simplify.

Line 281: should be the derivative melting curve

Line 375: add almost the same mobility as the protein

Line 405: correct FAM-TZD16/HAS with FAM-TZD16/HSA

Line 416: substitute the very promising with a very promising

Line 420; add “(apurinic/apyrimidinic) DNA” after AP

Line 455; correct with “the two observed products”

Line 437: add “the” after “each of”

Line 455: I would rephrase with: “For Ku antigen, two products corresponding to Ku70 and Ku80 subunits cross-linked with oligonucleotide were observed.

Line 467: I would rephrase with: “Using the radioactively labeled DNA allows for the detection of complexes at concentrations lower than the micromolar level, thus minimizing unspecific binding detection.”

Line 471: replace study with studies

Line 476: the bands of the oligonucleotide complexes with HSA and Ku seem resolved enough.

Line 477: add “oligonucleotides” after modified

Line 478: add “of” after complexes

Line 479: correct “smearing the bands” with the “smearing of the bands”

Line 480: replace “Either” with “Both”

Figure 6: line 9 has the wrong symbols: should be + in the HSA entry and – in the Ku entry.

Author Response

Thank you for the valuable suggestions and comments. We have carefully examined the comments and suggestions and revised the manuscript accordingly. We presented the word file with track changes. Please find as follows the responses to the comments. Please note that all the comments are bold-faced, and the authors' reply follows immediately below the comments.

The introduction is well developed and consistent with the topic of the manuscript, nevertheless the beneficial conjugation of lipophilic moieties as bile acids to oligonucleotides, should be considered  (Int. J. Mol. Sci. 2022, 23, 4270. https://doi.org/10.3390/ijms23084270)

Thank you for your suggestion. We have inserted the reference.

Line 255: the expression rationalize should be substituted with simplify.

Line 281: should be the derivative melting curve

Line 375: add almost the same mobility as the protein

Line 405: correct FAM-TZD16/HAS with FAM-TZD16/HSA

Line 416: substitute the very promising with a very promising

Line 420; add “(apurinic/apyrimidinic) DNA” after AP

Line 455; correct with “the two observed products”

Line 437: add “the” after “each of”

Line 455: I would rephrase with: “For Ku antigen, two products corresponding to Ku70 and Ku80 subunits cross-linked with oligonucleotide were observed.

Line 467: I would rephrase with: “Using the radioactively labeled DNA allows for the detection of complexes at concentrations lower than the micromolar level, thus minimizing unspecific binding detection.”

Line 471: replace study with studies

Line 476: the bands of the oligonucleotide complexes with HSA and Ku seem resolved enough.

Line 477: add “oligonucleotides” after modified

Line 478: add “of” after complexes

Line 479: correct “smearing the bands” with the “smearing of the bands”

Line 480: replace “Either” with “Both”

Figure 6: line 9 has the wrong symbols: should be + in the HSA entry and – in the Ku entry.

 Thank you for your suggestion. We have done the corrections in the text.

Reviewer 2 Report

Comments and Suggestions for Authors

The manuscript written by Bauer et al. is about studying the potential of oligonucleotide conjugates modified with a triazinyl phosphoramidate modification containing two dodecyl groups with the ability to self-assemble into micelles. The authors have carried out a systematic study that involves the biophysical characterization of such micelles in terms of dynamic light scattering, the critical aggregation concentration determination, and their ability to interact with human serum albumin (HSA). Additionally, the authors demonstrate the ability of these oligonucleotide conjugates to impart cellular uptake in two cell lines (HEK293 and T98G). This manuscript appears to be a continuation of an article previously published in Russ. J. Bioorg. Chem. 2021, 47, 719-733. The manuscript is interesting, well-written, and well discussed. The data is convincing with appropriate control experiments.

I have some comments and questions.

1.  The authors should include the MALDI-TOFF mass spectra off all oligonucleotide conjugates listed on Fig. 1A. Additionally, a table showing the mass (calculated) and mass (observed) is required.

2. Figure2. It seems that the behavior of duplex FAM-TZD17/TAMRA-M27 is quite different from their FAM-TZD16/M16 and TZD30/M30-FAM counterparts in terms of interactions with HSA, according to the electrophoretic mobility shift assay. Do the authors have an explanation of this result?

3. Lines 389-392. The authors suggest that “duplexes with shorter sequences more easily melt, causing the complementary chain to displace”, as a final statement. This is true, however, the melting temperature of FAM-TZD17/TAMRA-M27 is 59.9 ± 0.1 oC. During the binding experiment with HSA and EMSA assay, the temperature does not exceed the melting point of the duplex and the authors’ hypothesis might not be the case. Could the authors explain better this statement? 

4. Cytotoxicity. Although the authors have carried out cellular uptake experiments with TZD30 and TZD30/M30-FAM, the cytotoxicity of these oligonucleotides should be evaluated in both cell lines (HEK293 and T98G) as this cytotoxicity analysis has not been analyzed in other articles.

Comments on the Quality of English Language

Minor editing of English language required

Author Response

Thank you for the valuable suggestions and comments. We have carefully examined the comments and suggestions and revised the manuscript accordingly. We presented the word file with track changes. Please find as follows the responses to the comments. Please note that all the comments are bold-faced, and the authors' reply follows immediately below the comments.

  1. The authors should include the MALDI-TOFF mass spectra off all oligonucleotide conjugates listed on Fig. 1A. Additionally, a table showing the mass (calculated) and mass (observed) is required.

Thank you for your comment. We have inserted Figure S1 and Table S1 of MALDI-TOFF mass spectra and MW values for modified oligonucleotides.

  1. Figure2. It seems that the behavior of duplex FAM-TZD17/TAMRA-M27 is quite different from their FAM-TZD16/M16 and TZD30/M30-FAM counterparts in terms of interactions with HSA, according to the electrophoretic mobility shift assay. Do the authors have an explanation of this result?
  2. Lines 389-392. The authors suggest that “duplexes with shorter sequences more easily melt, causing the complementary chain to displace”, as a final statement. This is true, however, the melting temperature of FAM-TZD17/TAMRA-M27 is 59.9 ± 0.1 oC. During the binding experiment with HSA and EMSA assay, the temperature does not exceed the melting point of the duplex and the authors’ hypothesis might not be the case. Could the authors explain better this statement? 

We were interested to compare the micelle size of oligonucleotide conjugates bearing dodecyl groups, but differing in the length (16-, 17- and 30-mer). We hypothesized that FAM-TZD17 and FAM-TZD16 conjugates can form secondary structures: the sequence of FAM-TZD16 may form stable hairpin (Figure S6), which was observed by thermal denaturation (Figure S2A), and the sequence of FAM-TZD17 may form hairpin and partial self-dimer (Figure S6). Interestingly, the nucleotide sequence of FAM-TZD17 also enables a formation of self-dimer of eight nucleotide bases (Figure S6), which may contribute to the formation of larger particles and/or to accelerate their aggregation and influence on their interaction with HSA. In addition, the FAM-TZD17/TAMRA-M27 forms a duplex with an overhanging single-stranded moiety (unlike FAM-TZD16/M16 and TZD30/M30-FAM duplexes), which may affect the interaction with albumin. Here, we only briefly discussed the effects of the oligonucleotide length and sequence of the lipophilic oligonucleotide derivatives on the micelle size and interaction with HSA. This issue requires further studies on a wider series of oligonucleotides.

In the current work, we can only claim that no direct correlation between the size of micelle-like associates, their electroforetical mobility and length of the lipid oligonucleotide conjugates due to the additional impact of the nucleotide sequence. We hope that our research will serve as a base for future studies on a wider series of oligonucleotides tended to focus on the length, sequence and intramolecular structure of the lipid oligonucleotide conjugates. Under electrophoresis conditions, thermodynamic stability decreases, and this effect depends on the length of the resulting duplex. Given the fact that the electrophoretic analysis was performed at elevated temperature (37 °C), and under these conditions the more stable TZD30/M30-FAM duplex remains stable and binds to the protein. In the case of 16-mers and 17-mers, partial displacement of a chain, which does not contain lipophilic residues, is observed. Finally, this effect may be associated with the formation of intramolecular structures by FAM-TZD17 and FAM-TZD16 oligonucleotides.

  1. Cytotoxicity. Although the authors have carried out cellular uptake experiments with TZD30 and TZD30/M30-FAM, the cytotoxicity of these oligonucleotides should be evaluated in both cell lines (HEK293 and T98G) as this cytotoxicity analysis has not been analyzed in other articles.

Thank you for your comment. We have inserted Figure S11 of the cytotoxicity experiments.

Round 2

Reviewer 2 Report

Comments and Suggestions for Authors

The authors have addressed all the reviewers' comments and therefore I recommend this manuscript for publication in Pharmaceutics

Comments on the Quality of English Language

Minor editing of English language required